# Estimation of long-term river discharge and benefits evaluation of ecological water conveyance in Ertanggou basin from 2000 to 2020

**Junqing Lei**[1], **Adilai Wufu**[1,2*], **Chao Xu**[1,2], **Zhanbo Wang**[1]

**1** Department of College of Resources and Environment, Xinjiang Agricultural University, Urumqi, China,
**2** Xinjiang Engineering Technology Research Center of Soil Big Data, Urumqi, China

\* Adilagupur@126.com

## Abstract

Given the current ecological degradation in the inland arid river basins of our country, ecological water conveyance has increasingly become a critical measure to alleviate the pressures of water scarcity on ecosystems and to promote regional sustainable development. The Ertanggou watershed in Xinjiang was selected as the study area. This study utilized Remote Sensing Hydrological Station (RSHS) technology to calculate river discharge for typical cross-sections in the Tulufan Ertanggou basin from 2000 to 2020. The study also analyzed the impact of ecological water conveyance on different land types using land cover type data to assess the benefits of ecological water conveyance. RSHS technology offers reliable data support for estimating river discharge and evaluating ecological water conveyance. The average annual growth rates of the river basin discharge at the study sections are 0.36%, 1.2%, and 0.77% respectively from 2000–2016. All monitored river sections were glacial meltwater-dominated, and the accelerated melting of glaciers caused by climate warming was the main reason for the increasing trend. From 2016 to 2020, ecological water conveyance provided an average of $0.61 \times 10^8$ m³ of ecological water per year, with up to 10% water conveyance efficiency. Reservoir development and river training have significantly increased this efficiency. The ecological water supply was significantly correlated with changes in landscape patterns, such as cultivated land, grassland, and water bodies in the downstream area. The water area expanded by $6.14 \times 10^5$ m², while cultivated land increased by 32.98%. It is evident that ecological water conveyance effectively promotes land use optimization and ecological balance. This study will provide new ideas for evaluating the benefits of ecological water conveyance in inland arid basins with limited data, offering theoretical support for future ecological water conveyance projects.

**Data availability statement:** All relevant data are within the paper and its Supporting Information files.

**Funding:** The corresponding author, Adilai Wufu (A.W.), received financial support from the Xingjiang Uighur Autonomous Region Talent Project ("Tianchi Yingcai" Talent Introduction Program 2022). The funding institution is the Human Resources and Social Security Department of Xinjiang Uighur Autonomous Region. The author Xu Chao (X.C.) was funded by the Science and Technology Department of Xinjiang Uygur Autonomous Region (Project No.2023A02002-4). The funding institution is the Science and Technology Department of Xinjiang Uighur Autonomous Region.

**Competing interests:** The authors have declared that no competing interests exist.

# 1. Introduction

Water is the source of life, maintaining human survival and development and supporting all living things and ecosystems on Earth [1]. With population growth and economic development, the demand for water resources is also increasing. However, water resources are not infinite, and freshwater resources are minimal. Therefore, the rational use and protection of water resources have become an important global issue. River discharge is a vital freshwater resource and a key indicator for measuring the abundance of surface water resources. In recent years, numerous studies have been conducted on river discharge's changing patterns and causes. For example, Li Lijuan et al. [2] found that human activities have become the main driving factor in the evolution of rivers in the Chaobai River Basin. Xie Hongxia et al. [3] analyzed the impact of land use and climate change on discharge in Hunan Province's Zhengshui Basin from 1990 to 2010. They found that climate change primarily influenced discharge changes, with land use changes contributing relatively less. Huang Binbin et al. [4] taking the water cycle process as the main line, clarified that there are significant regional differences in the contribution of climate change and human activities on surface discharge changes among different river basins worldwide. The changes in river discharge and cause analysis have received widespread attention. Still, due to the influence of river genesis—surface water flows down from the cracks and folds in the ground according to topography, erodes the ground for a long time, scours into gullies, forms streams, and finally converges into rivers. Reducing river water consumption and maximizing its use by downstream residents have become a more profound problem for the general public [5]. In arid regions, in particular, due to their inland location, distance from the ocean, and the barrier of high mountains, it is difficult for moist air currents from the sea to reach, resulting in scarce precipitation [6,7]. At the same time, due to the influence of extremely high temperatures and intense sunlight, regional evaporation is far greater than precipitation. This unbalanced water balance between income and expenditure exacerbates drought and makes water resources even more scarce. As the standard method of water replenishment in inland arid basins, ecological water conveyance transports water resources to downstream areas with scarce water in a particular scale, time, and manner, playing an essential role in maintaining and restoring ecosystem structure and function, as well as promoting ecosystem balance [8]. Ecological water conveyance directly affects land use changes in arid areas, particularly in areas where agriculture is the primary socioeconomic source, and the amount of ecological water conveyance significantly impacts residents. Therefore, evaluating ecological water conveyance benefits and quantifying its impact on land use types provide a scientific basis for formulating land use policies and water resource management strategies, which is crucial for maintaining downstream ecological security [9].

At present, numerous scholars have conducted extensive research on ecological water conveyance monitoring and its benefit evaluation. For instance, some scholars have analyzed the benefits of ecological water conveyance in the lower reaches of the Tarim River basin using multi-source remote sensing data, Exploring the positive impact of ecological water conveyance on the basin's ecological

environment [10,11]. Wang Chuan et al. [12] conducted an analysis of the effects of ecological water conveyances on groundwater, surface water, and vegetation indices within the Ejina Oasis, located in the lower reaches of the Heihe River. Their study utilized long-term monitoring and satellite remote sensing data to assess the impact comprehensively, emphasizing the crucial role of ecological water conveyance in improving the hydrological environment and promoting vegetation recovery in arid regions. Dong Zhiling et al. [13] studied the impact of ecological water conveyance, specifically focusing on its effects on downstream vegetation in the Shiyang River terminal lake basin. They found that implementing artificial measures to assist vegetation recovery during ecological water conveyance can enhance species richness. Extensive research has been done on the benefits of ecological water conveyance. However, there are still challenges in combining river discharge with the benefits of ecological water conveyance and determining its impact on agricultural land. This is mainly because river discharge, a critical factor in evaluating the ecological benefits of water conveyance, is influenced by complex geographic conditions, climate, transportation, and other factors. Many areas worldwide have not been observed or lack observed information, making it extremely difficult to obtain discharge data [14,15]. Additionally, previous research often focuses solely on theoretical aspects, lacking practical application. There is a need to strengthen the integration with practice to provide a more scientific basis for the rational utilization and management of water resources.

Our research aimed to estimate the long-term river discharge in the Tulufan Ertanggou basin from 2000 to 2020 and analyze the benefits of ecological water conveyance. We focused on two main issues: (1) Estimating long-term river discharge using remote sensing hydrological station method. (2) Quantitatively analyzing the impact of ecological water conveyance on different land cover types and evaluating the associated benefits. We aim to provide a scientific basis for water resources planning and constructing water conservancy projects in the Ertanggou basin. Additionally, we hope to offer insights and references for downstream agricultural development in this region.

## 2. Materials and methods

### 2.1. Study area

The Ertanggou basin (89°54′~89°57′E, 43°16′~43°19′N) is located in the arid region of northwest China, in Shanshan County, Turpan City, Xinjiang Uygur Autonomous Region (Fig 1). The river mainly originates from the southern foot of the Bogda Mountain of the Tianshan Mountains and is a typical mountain-stream inland river [16]. High mountains surround the basin, and the ocean air is not easy to reach, resulting in a typical temperate continental arid climate with long sunshine hours and significant temperature differences. The average annual temperature is 14.5 ℃, with the average temperature in summer reaching as high as 35 ℃, and the annual rainfall is very scarce, only 17 mm. In recent years, water resource shortages in the Shanshan County basin have led to significant challenges, including river interruptions, vegetation degradation, and difficulties in securing adequate irrigation water for farmlands in downstream river channels and residential areas [17]. To address the pressing demands of economic and social development and to optimize the allocation of water resources within the basin, the government completed the "Feasibility Study Report on Key Water Resource Allocation Projects in Shanshan County" in 2015. Following this, a series of water-saving transformation measures were implemented. Among these efforts, the construction of the Ertanggou Reservoir began in September 2012. By November 2014, the reservoir was operational, with a total storage capacity of $2.36 \times 10^8$ m³ and an irrigation control area extending to 202,600 acres. This reservoir has played a crucial role in alleviating severe water shortages in downstream oasis communities, increasing irrigation water for agricultural fields, and providing a reliable water resource supply for the daily needs of downstream urban residents [18].

By August 2016, the Ertanggou Reservoir had reached its normal storage level. It is located at the third cross-section, which was formed following the completion of river channel regulation and alignment works. However, the absence of suitable hydrological stations makes it difficult to monitor changes in river flow prior to the implementation of ecological water conveyance.

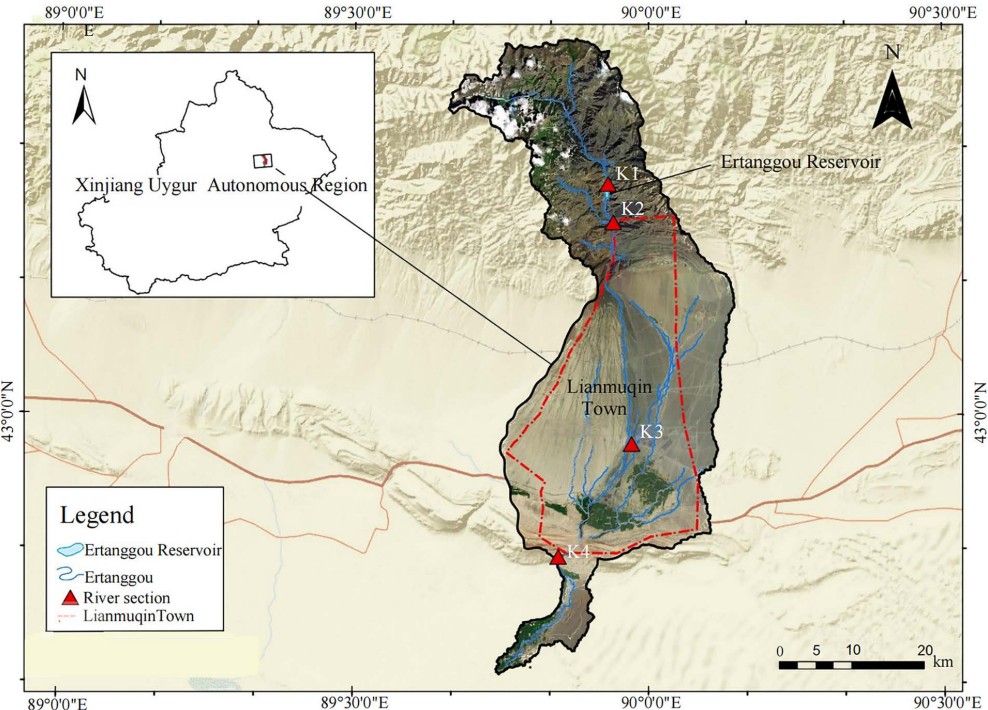

**Fig 1. The Ertanggou watershed and locations of monitoring sections.**

## 2.2. Datasets

This study mainly used three types of data: remote sensing data, field survey data, and land cover type data. The data information and uses are shown in Table 1.

### 2.2.1. Unmanned aerial vehicles (UAV) and field measurement data.
Low-altitude remote sensing imagery was obtained using a DJI Phantom 4 Professional UAV (Dajiang Company, Shenzhen, China, https://www.dji.com/cn.) between 9 and 12 August 2023. The DJI 4 was used to fly on a fixed route to obtain high-definition image data. An action camera (FC300X) was mounted on the Phantom 4, which can shoot 4K video or 12-megapixel stills. After the acquisition of the data, laboratory data processing was undertaken. This included data preprocessing, point cloud encryption through the Pix4D software (https://pix4d.com/. Accessed on 30 June 2020), generation of the high-precision digital orthophoto map (DOM), and generation of the digital surface model (DSM). The essential hydrological data for discharge estimation were measured during the UAV flight, including flow velocity, river discharge, water depth, and underwater topography. The flow velocity was measured using a surface velocity radar (SVR, USA), the river discharge was measured using a portable Ponolflow-VA Doppler flow instrument, and water depth was measured using a staff gauge to survey. These measured data provide input data for the discharge calculation formula and were used to validate the estimation results.

### 2.2.2. Satellite remote sensing data.
The river surface width for the 2000–2020 monthly scale was determined using open and accessible satellite imagery, including Sentinel-2 series and Landsat series images as data sources. Using the Google Earth Engine (GEE, https://earthengine.google.com), conducting image projection, image stitching, cutting, and radiation correction calibration, and the normalized differential water index (NDWI) is calculated by using green and near-infrared bands to calculate the water area of each month.

$$NDWI = (\rho_{GREEN} - \rho_{NIR})/(\rho_{GREEN} + \rho_{NIR}) \tag{1}$$

**Table 1. Data information and use.**

| Type | Definition | Source | Period | Spatial Resolution | Goal |
|------|-----------|--------|--------|-------------------|------|
| Low-altitude remote sensing data | UAV remote sensing data | DJI Mavic Air 2 | 2022/8-20022/9 | 5 cm | Construct a digital channel model |
| Satellite remote sensing data | Landsat-5 Landsat-7 Landsat-8 Sentinel-2 | GEE | 2000/3-2011/10 2012/10-2013/03 2013/04-2015/10 2016/03-2020/10 | 30 m 30 m 30 m 10 m | Identify the water body and extract the river width |
| Ground measured data | Flow rate data Water depth data | Observation and measurement in situ | 2022/8-2022/9 | / | Build RSHS |
| Satellite remote sensing data | Landsat-7 | Resources and Environmental Science Data Center | 2000-2020 | 30 m | Calculate the land-use area |

where $\rho_{GREEN}$: the reflectance value of the green light band in remote sensing image; $\rho_{NIR}$: the reflectance value of the near infrared band in remote sensing image.

**2.2.3. Land cover type data.** To ensure data consistency, we used land cover type data from three time periods (2000, 2010, and 2015) obtained from the Data Center for Resources and Environmental Sciences of the Chinese Academy of Sciences. The data was derived from Landsat TM/ETM remote sensing images and was used to create a 1:100,000 scale land use grid map. The land cover types were classified into six groups based on local environmental characteristics: cultivated land, forest land, grassland, water area, construction land, and unused land.

## 2.3. Methods

**2.3.1. Flow estimation method using remote sensing hydrographic station.** To gather river discharge data in the study region, we used remote sensing hydrology station (RSHS) technology to estimate the flow of four typical river sections in the Ertanggou Watershed [19,20]. RSHS technology is a method that combines low-altitude remote sensing data with field-measured data. It then simulates and analyzes hydrological information based on hydrology, hydraulics, and quantitative remote sensing. This method constructs a small and medium-sized river model in areas without measurement stations and estimates flow using remote sensing [21]. RSHS technology is currently used in various fields, and its outcomes are promising. The system's river flow data can efficiently supplement discharge data in places without hydrographic stations or with limited hydrological data and provide data assistance for water resource management in desert areas [22]. The exact algorithm of RSHS technology consists of the following three steps, and the application process is as follows:

**2.3.1.1. Construction of digital river model:** We conducted UAV observations and field measurements to collect precise data on UAV and relevant hydraulic parameters, such as river depth and velocity. The Pix4d software processed the UAV images, created a digital orthophoto image (DOM) and a digital surface model (DSM), and then merged the measured river depth and velocity to produce a 3D digital river model.

**2.3.1.2. Retrieving the river's width using satellite remote sensing:** The slope and roughness were assessed through UAV imagery and field measurement outcomes, while the width of the water surface was measured concurrently. We extracted a specified river segment's monthly water surface areas by integrating historical satellite remote sensing data with sub-pixel decomposition techniques. This has been achieved through the mass download and analysis of NDWI remote sensing imagery. The river width was then determined by calculating the ratio of the water surface area to the length of the river segment. The formula used for this calculation is as follows:

$$W = A_{water}/L_{water} \tag{2}$$

where W: the average water surface width (m); $A_{water}$: the water surface area of the reach ($m^2$); $L_{water}$: the length of the river (m).

**2.3.1.3. Estimating river discharge in long time series:** The digital river model estimates the river section's water depth, water level, overcurrent area, wet circumference, and hydraulic radius at different times. Then, the Manning formula is applied to calculate the flow velocity. Finally, the river discharge for various historical periods is determined by multiplying the flow velocity by the flow area. The monthly average is obtained from remotely sensing inversion and multiplied by time to calculate the monthly scale discharge. The computation formula is provided below.

$$R=A/L \tag{3}$$

$$V=\frac{k}{n}\times R^{2/3}\times J^{1/2} \tag{4}$$

$$Q=V\times A=\frac{k}{n}\times A^{5/3}\times J^{1/2}\div L^{2/3} \tag{5}$$

where V: flow velocity (m/s); k: conversion constant, k = 1; n: roughness; R: hydraulic radius (m); J: gradient; A: flow area ($m^2$); L: wet circumference (m); Q: river discharge ($m^3$/s).

**2.3.2. Accuracy assessment.** The accuracy of the remote sensing inversion flow method was tested to assess its reliability. The validation focused on estimating river cross-section flow velocity and instantaneous discharge using UAV and Manning's formula. The estimation results were evaluated using statistical metrics such as relative accuracy (RA), root mean square error (RMSE), and the Nash-Sutcliffe efficiency coefficient (NSE) [23]. The computation formula is provided below.

$$RA=\frac{|Q_c-Q_a|}{Q_a} \tag{6}$$

$$RMSE=\sqrt{[\frac{\sum (Q_c-Q_m)}{n}]^2} \tag{7}$$

$$NSE=1-\frac{\sum_{t=1}^{T}(Q_m^t-Q_c^t)^2}{\sum_{t=1}^{T}(Q_m^t-\overline{Q_m})^2} \tag{8}$$

where T: the total number of periods; $Q_m^t$: the measured discharge ($m^3$/s) during the period t; $Q_c^t$: the estimated discharge ($m^3$/s) at the period t; $Q_m$: the average measured discharge ($m^3$/s) over the T period.

**2.3.3. Evaluation of watershed discharge effects.** The discharge effect was analyzed using two criteria: ecological water conveyance and water conveyance efficiency, land landscape pattern change and land use area change. Ecological water conveyance refers to the total water entering the downstream river yearly. In contrast, water conveyance efficiency refers to the ratio of total water reaching Lianmuqin Town to ecological water conveyance each year (Table 2). The change in land landscape pattern is the change in the underlying land cover and land use type in the spatial distribution, configuration, shape, and size caused by ecological water conveyance. This is primarily analyzed by the change of natural land cover type and land use mode. Land use area change can reflect the efficiency and intensity of land use. The land use area conveyance matrix objectively assesses the trend and pattern of land use change.

**2.3.4. Pearson correlation analysis.** Pearson correlation coefficient analysis was used to measure the degree of tight association between ecological water conveyance and land use area during the study period. The method helped reflect the intensity of the watershed's dynamic response to surface discharge. A more excellent correlation coefficient

**Table 2. Evaluation data of discharge effect.**

| Data name | Design formulas | Remark |
|---|---|---|
| Ecological water conveyance | $W_{1-eco} = \sum_{t=1}^{12} Q_1^t \times 3600 \times 24 \times 30$ | $W_{1-eco}$: the total water volume reached in the K1 section in one year(m³); $Q_1^t$: the mean monthly discharge of the t-month K1 section(m³/s); |
| Lianmuqin town water reserves | $W_{2-eco} = \sum_{t=1}^{12} Q_3^t \times 3600 \times 24 \times 30$ | $W_{2-eco}$: the total water volume has arrived in Lianmuqin town in one year (m³); $Q_3^t$: the mean monthly discharge of the t-month K3 section(m³/s); |
| Water conveyance efficiency | $\tau = \dfrac{W_{2-eco}}{W_{1-eco}} \times 100$ | $\tau$: the water conveyance efficiency of the year (%); |

value indicates a stronger correlation between the variables, while a lesser value indicates a weaker connection. The computation formula is provided below.

$$r = \frac{\text{cov}(X,Y)}{\partial_X \partial_Y} \tag{8}$$

where cov: covariance; $\partial$: standard deviation;

## 3. Results and analysis

### 3.1. Changes in river discharge

**3.1.1. Digital river modeling of study sections.** After conducting a comprehensive field assessment of the Ertanggou Watershed and its surrounding environment, we surveyed the high-altitude mountain river using UAV monitoring. We selected four specific areas for detailed analysis. To create DOM (Digital Orthophoto Map) and DSM (Digital Surface Model) images, we preprocessed the UAV data using the professional software Pix4Dmapper (https://pix4d.com), which involved splicing and encrypting point cloud data (Fig 2). Our findings show that DOM can identify riverbank boundaries, riverbed environments, and nearby plant life. Additionally, DSM can accurately capture terrain details such as cross sections and profiles above the water's surface.

The elevation changes in the terrain were obtained using high-precision DSM data captured by a UAV along with the ArcGIS spatial analysis tool. The middle section of the river, as captured by the uncrewed aerial vehicle (UAV), is selected as the cross-section for flow estimation, ensuring that the upstream and downstream distances are approximately equal. For the submerged portions of the riverbed, underwater topography is acquired through direct cross-sectional measurements. Meanwhile, the topography above the water surface is derived from DSM data obtained via UAV surveys. By integrating the UAV-captured terrain data for the above-water portion with the measured underwater data, the complete cross-sectional profile of the river channel is generated through seamless splicing of the upper and lower terrains. Based on this information and the field-collected data, the digital model of the river for each monitoring segment was developed using the RSHS computation tool (Fig 3). The digital river channel model on the left effectively represents the three-dimensional structure of the river channel, providing a clear depiction of its shape. In comparison, the cross-sectional profile graph on the right illustrates the water depth and surface width at the selected cross-section. The water level variations simulated by the model align closely with trends observed during field surveys, demonstrating the model's capability to accurately capture the hydrological processes of the studied river section.

**3.1.2. Evaluate the accuracy of remote sensing inversion of instantaneous flows.** The RSHS approach was utilized to estimate the instantaneous discharge at four typical river crossings in the basin. This estimation was validated by comparing it with field-measured flow velocity and river discharge (Table 3). The results showed that the error standard for velocity and discharge is less than 20%, which aligns with the "Hydrological Information Forecast Code" provided by China's water conservancy agency. The accuracy of the estimation was assessed using Eqs. (7) and (8), resulting in RMSE values of 0.2 m/s, 3.2 m³/s, and NSE values of 0.97 and 0.94, respectively. These results demonstrate that using

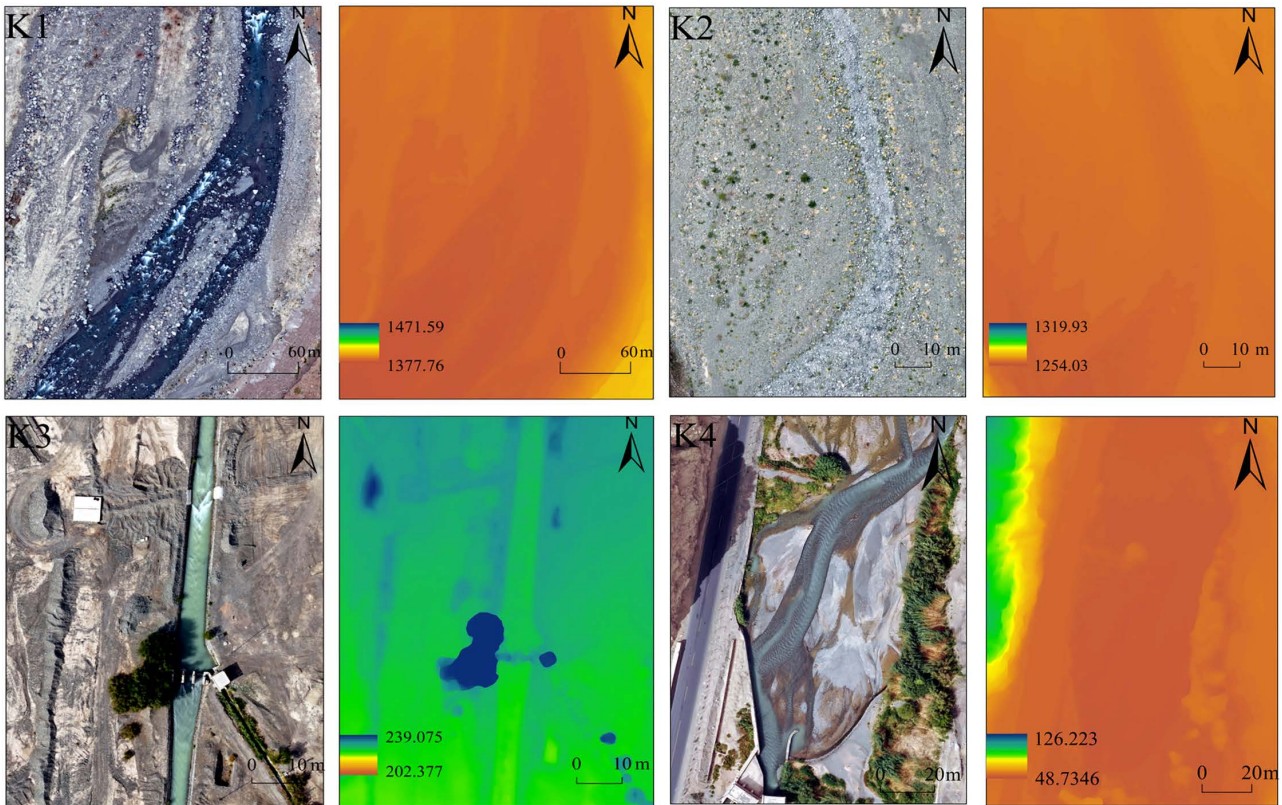

**Fig 2. The DOM and DSM plots of the study sections.**

RSHS technology for river flow velocity and discharge estimation is satisfactory, meets accuracy criteria, and is highly reliable.

**3.1.3. Analysis of long-term river discharge change.** Based on GEE's monthly-scale NDWI data from the 2000–2015 Landsat series and the 2016–2020 Sentinel-2 series, the RSHS was used to estimate river discharge data for cross-sections K1, K2, and K4 for 2000–2020, and for cross-section K3 for March to November 2016–2020 (Fig 4). The time from December to February is excluded as it falls within the glacial period. The area where K3 is located is part of the Ertanggou Artificial Canal, which began operations in November 2014, with river control and consolidation activities ongoing in 2015. Therefore, monthly discharge estimating data for the K3 segment started in 2016. As shown in the figure, between 2000 and 2016, the average annual growth rates of discharge at the K1, K2, and K4 cross-sections were 0.36%, 1.2%, and 0.77%, respectively. This upward trend can be primarily attributed to the accelerated melting of glaciers driven by global warming [24]. In particular, the increased contribution of glacial and snowmelt water from the Tianshan Bogda Mountain area—the primary water source of the Ertanggou Basin—has led to a rise in water input into the basin. This phenomenon has also caused seasonal variations in discharge, with the highest rates occurring during the summer when elevated temperatures accelerate glacier melt, resulting in peak discharge.

From 2016 to 2020, the annual discharge at the four cross-sections exhibited a marked overall increase, with average annual growth rates of 1.34%, 3.14%, 3.13%, and 7.71%, respectively. A comparative analysis of the two periods (2000–2016 and 2016–2020) reveals that the average annual growth rates of discharge at the K1, K2, and K4 cross-sections significantly increased after 2016. This trend underscores the substantial role of reservoir operations post-2016 in enhancing cross-sectional discharge.

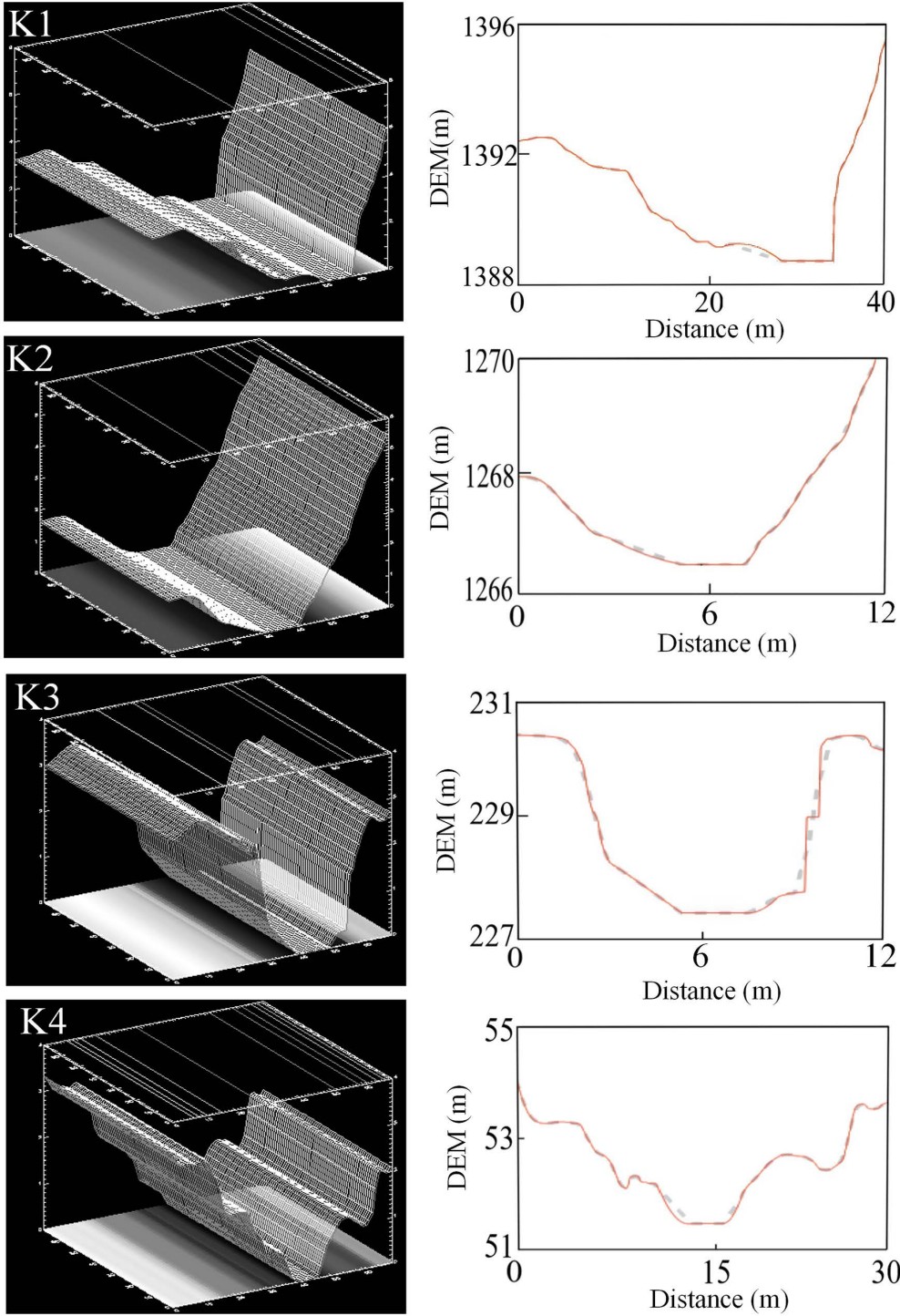

**Fig 3. River cross-section and digital channel model diagram.**

**Table 3. Relative accuracy evaluation table of flow velocity and discharge.**

| Section | Measured flow velocity (m/s) | Estimate flow velocity (m/s) | RA (%) | Section | Measured discharge (m³/s) | Estimate discharge (m³/s) | RA (%) |
|---------|------------------------------|------------------------------|--------|---------|---------------------------|---------------------------|--------|
| K1 | 1.35 | 1.57 | 16.3 | K1 | 32.1 | 25.76 | 19.8 |
| K2 | 4.1 | 3.92 | 4.4 | K2 | 2.8 | 2.3 | 17.9 |
| K3 | 2.1 | 1.86 | 11.4 | K3 | 2.2 | 1.92 | 12.7 |
| K4 | 1.15 | 1.31 | 13.9 | K4 | 2.35 | 2.71 | 15.3 |

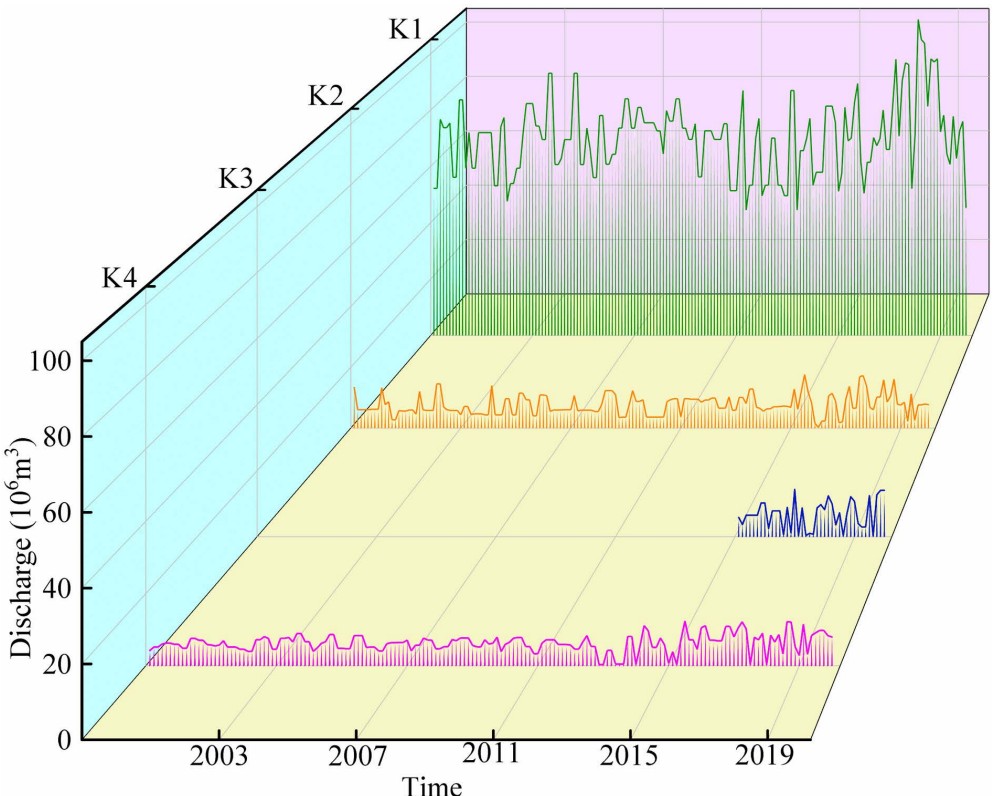

**Fig 4. The changes in long-term series river discharge.**

In July 2018, cross-sectional discharge peaked, primarily due to the impact of climate change. During this period, temperatures in Xinjiang were higher than in previous years, leading to accelerated glacier melt and a considerable increase in flow. However, climate change remains subject to seasonal effects, causing a pronounced peak during summer followed by a gradual decline [25].

Between 2016 and 2020, the monthly discharge at the K1 cross-section was significantly higher than at the other cross-sections, nearly three times greater. This can be attributed to the construction and operation of the Ertanggou Reservoir, which achieved the goal of river interception, effectively regulated downstream flow, ensured sufficient water availability for downstream ecological needs, and maintained the stability of the river ecosystem.

Over the past 20 years, the annual discharge at the K1 cross-section has consistently exceeded that of the downstream cross-sections. This disparity is largely due to geographic factors, as the K1 cross-section is situated at the

mountain outlet upstream of the reservoir at a higher elevation, where glacial meltwater directly contributes to river flow. In contrast, as elevation decreases downstream, surface discharge slows, and water flow becomes more dispersed. Additionally, the downstream region, characterized by dense populations and intensive agricultural activity, diverts substantial water resources for irrigation, industrial, and domestic use, further reducing water reserves.

Since 2016, following the completion of river channel regularization projects, the monthly discharge at the downstream K2, K3, and K4 cross-sections has shown a slight declining trend. However, the fluctuations have been minimal, with discharge levels remaining relatively stable. This demonstrates that the implementation and optimization of water conservancy projects can significantly reduce water loss during conveyance and increase the volume of water available for downstream use.

### 3.2. Assessment of basin discharge effects and implications

#### 3.2.1. Changes in ecological water conveyance and water conveyance efficiency.
The discharge data from cross-sections K1 and K4, estimated using the RSHS method, were used to evaluate the discharge effects (Table 2) and calculate the amount of water conveyance and efficiency in the basin (Fig 5). From 2000 to 2013, the discharge of K1 and K4 sections remained relatively consistent, with average annual discharges of $5.45 \times 10^8$ m³ and $0.48 \times 10^8$ m³, respectively. The ecological water conveyance efficiency during this period was approximately 8%. However, in 2014, despite slight variation in the discharge of the K1 section compared to previous years, the watershed's ecological water conveyance efficiency dropped to 4.8%. The construction of the reservoir has been the primary factor influencing this trend. In 2014, during the construction of the Ertanggou Reservoir and the downstream river channel regularization and rectification, a significant amount of ecological water was lost, leading to a substantial reduction in the water volume reaching the K4 section. However, between 2015 and 2020, following the completion of the reservoir and artificial channel

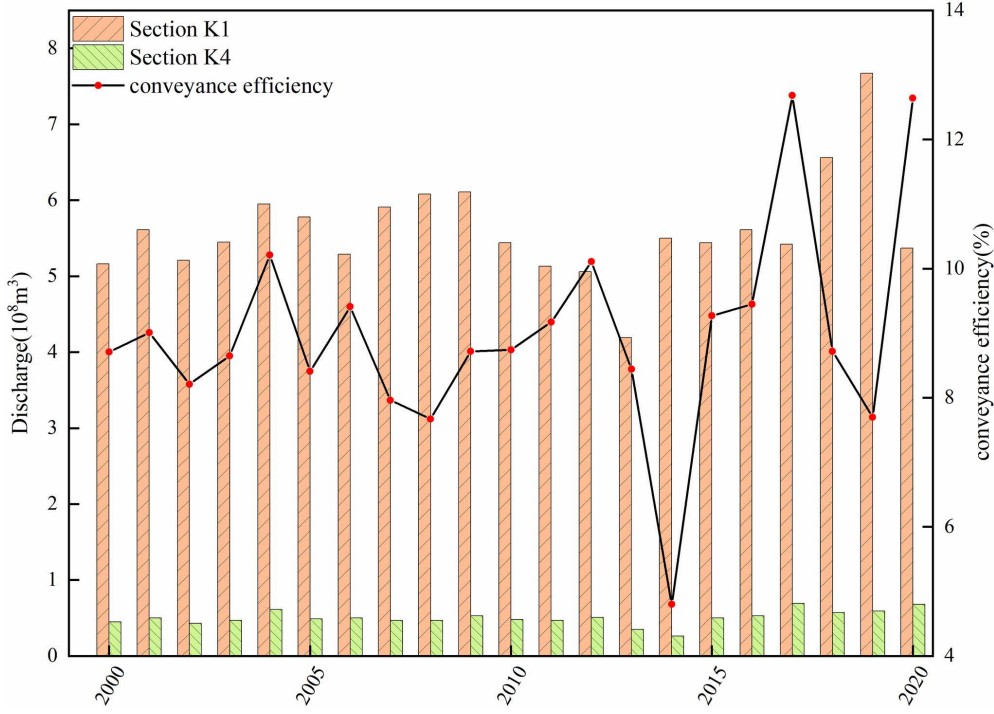

**Fig 5. Ecological water conveyance and water conveyance efficiency.**

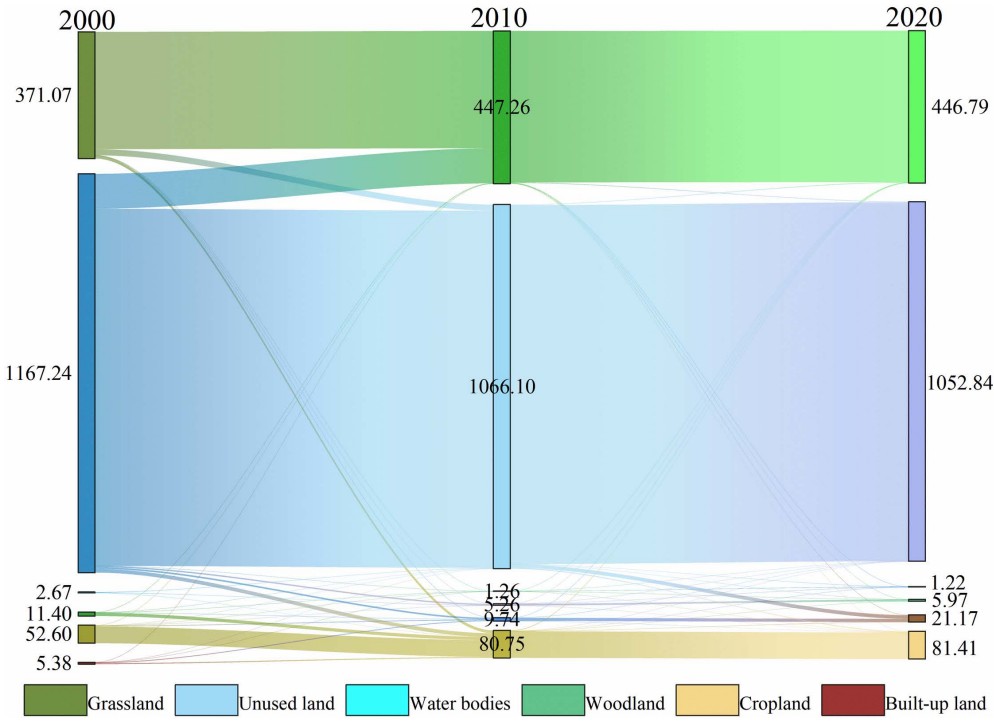 PLOS One

construction in November 2014 and the gradual improvement of river channel regularization efforts in 2015, water resources were allocated and utilized more effectively.

During this period, the discharge at the two sections exhibited a statistically significant increasing trend (P<0.05). Furthermore, the average annual ecological water conveyance volume rose to 0.61 × 10⁸ m³, while the water conveyance efficiency exceeded 10%.

In 2018, due to Xinjiang's arid climate, precipitation along the watershed decreased, leading to an overall reduction in water reserves compared to previous years [26]. Despite this, water conveyance efficiency might still reach approximately 8%. This fact indicates that implementing ecological water conveyance can increase the total water volume reaching the downstream area, improve flow conditions, reduce leakage and evaporation losses, optimize water resource allocation and utilization, and allow more water resources to efficiently get to the downstream residential oasis area while reducing ecological water conveyance loss and improving water use efficiency.

**3.2.2. Analysis of changes in landscape patterns in watersheds.** Fig 6 shows the Sankey diagram illustrating changes in different land use types in the Ertanggou Watershed from 2000 to 2020. It demonstrates that the landscape composition has changed to varying degrees over the last two decades. In addition to large areas of unused land, grassland has been the primary land use type in the Ertanggou Watershed. Built-up land is gradually expanding from the central location to the northeast of Lianmuqin Township, and the cropland area, the leading agricultural land, is also increasing yearly.

After obtaining the land use area and composition of the watershed, a statistical analysis of the data for the three periods of 2000, 2010, and 2020 (Table 4) was conducted to calculate the rate of change based on the amount of land use change.

The data reveals significant changes in the water bodies and built-up land between 2000 and 2020. The change rate for the water bodies is 877.14%, with an annual change rate of 43.86%. For built-up land, the change rate is 321.52%,

**Fig 6. Sanki diagram of landscape composition transfer in watershed 2000-2020.**

**Table 4. Land use types and changes in Ertanggou watershed from 2000 to 2020.**

| 2000 | 2000 | | 2010 | | 2020 | | 2000–2020 | |
|---|---|---|---|---|---|---|---|---|
| | Area (10⁵m²) | Percentage (%) | Area (10⁵m²) | Percentage (%) | Area (10⁵m²) | Percentage (%) | Change rate (%) | Annual gradient (%) |
| Grassland | 230.37 | 31.95 | 684.86 | 53.64 | 671.38 | 49.31 | 191.44 | 9.57 |
| Cropland | 383.52 | 53.20 | 487.23 | 38.16 | 509.99 | 37.48 | 32.98 | 1.65 |
| Built-up land | 40.28 | 5.58 | 92.11 | 7.21 | 169.79 | 3.05 | 321.52 | 16.08 |
| Woodland | 66.15 | 9.17 | 6.25 | 0.49 | 6.27 | 0.56 | -90.5 | -4.52 |
| Water bodies | 0.70 | 0.10 | 6.35 | 0.50 | 6.84 | 0.60 | 877.14 | 43.86 |

with an annual change rate of 16.08%. These changes are mainly attributed to the construction of the Ertanggou Reservoir, which has effectively controlled water flow and facilitated the industrialization and urbanization of Lianmuqin Township. Additionally, the area occupied by cropland and grassland has consistently increased and is a significant part of the watershed area. This growth is closely linked to the Xinjiang government's efforts to develop the western part of the country while prioritizing ecological protection. Conversely, forest land has experienced a change rate of -90.5%, primarily due to deforestation for agricultural expansion in Lianmuqin Township.

Land use change is a dynamic process that involves changes in both directions, including the conversion from one land category to another and the reversal process from other land categories back to the original one. The direction of land use change in the watershed was statistically analyzed using ArcGIS spatial overlay analysis, creating a land use transfer matrix (Table 5).

According to Table 5, a significant amount of unusable land around the watershed will be utilized between 2000 and 2020. Approximately 1/4 of the unusable land will be converted into grassland. A considerable portion of the idle land has been reclaimed or used for construction. For example, 1225.17 km² of unused land has been transformed into cropland, 1608.03 km² has been converted into built-up land, and 382 km² of unused land has been converted into water bodies. This change is mainly because Xinjiang, China's largest province, contains many deserts and Gobi, and land resources are limited. However, over the last 20 years, Xinjiang has strengthened ecological protection by constructing water conservancy facilities, advancing irrigation and water-saving technology, and continuously improving ecological environment quality. The national western development strategy has also led to many reclamations of irrigated grassland for farmland and increased built-up land.

**3.2.3. Analysis of driving factors for ecological water conveyance change in the watershed.** The Pearson correlation analysis method examined the relationship between land use type index and ecological water conveyance. The results are shown in Fig 7. The correlation values of ecological water conveyance with cropland, grassland, and water bodies are all greater than 0.6, indicating a significant positive correlation. However, the correlation value with woodland is -0.7, indicating a substantial negative correlation. This suggests that the increase in ecological water conveyance

**Table 5. Watershed land use transfer matrix table.**

| Type | Grassland | Cropland | Built-up land | Woodland | Water bodies | Unused land | Total |
|---|---|---|---|---|---|---|---|
| Grassland | 34316.64 | 825.03 | 7.38 | 75.42 | 81 | 1790.1 | 37095.57 |
| Cropland | 92.79 | 4887.18 | 218.34 | 0.81 | 1.26 | 57.78 | 5258.16 |
| Built-up land | 19.17 | 259.02 | 244.17 | 0.99 | / | 14.4 | 537.75 |
| Woodland | 36.9 | 945.63 | 39.33 | 41.13 | / | 76.68 | 1139.67 |
| Water bodies | / | 1.17 | / | / | 131.76 | 133.65 | 266.58 |
| Unused land | 10221.12 | 1225.17 | 1608.03 | 3.6 | 382.68 | 103244.31 | 116684.91 |
| Total | 44686.62 | 8143.2 | 2117.25 | 121.95 | 596.7 | 105316.92 | 160982.64 |

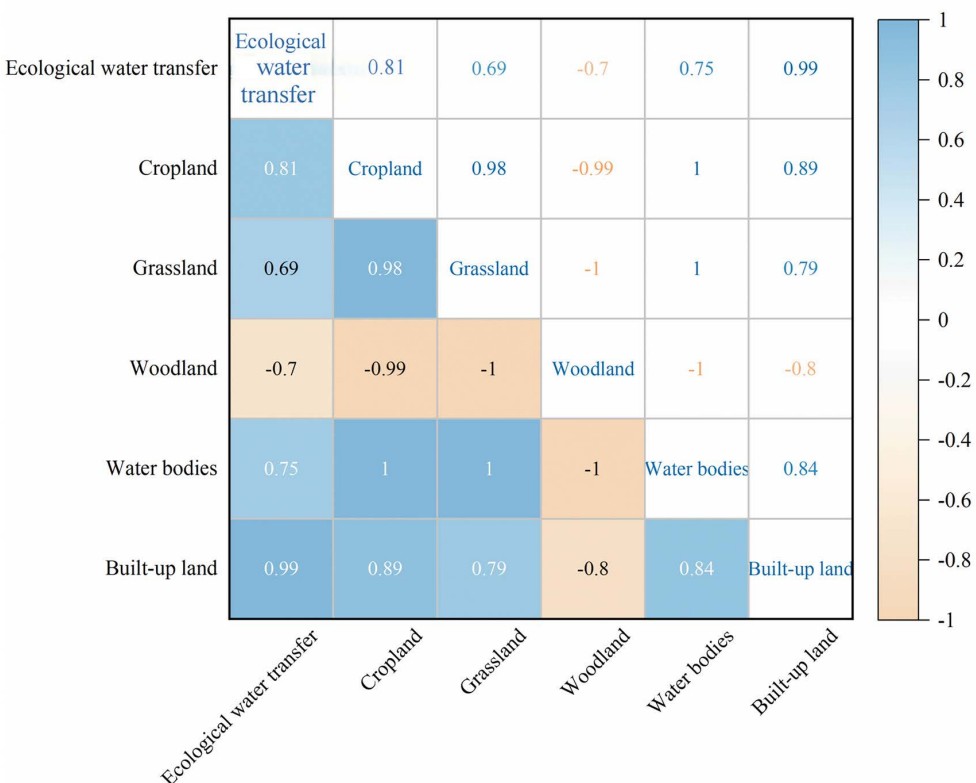

**Fig 7. Correlation analysis of each factor.**

is directly related to the growth of cropland, grassland, and water bodies between 2016 and 2020. Ecological water conveyance is essential for improving the water condition of cultivated land and increasing land utilization rates. It also helps improve the grassland's ecological environment and promotes vegetation recovery and growth.

Additionally, the development of the water bodies is positively correlated with ecological water conveyance, indicating that ecological water conveyance is vital for maintaining and restoring the ecological environment of the water area, which is crucial for ecosystem protection and improving water resource utilization efficiency. In summary, increasing ecological water conveyance effectively alleviates the problem of insufficient cropland and green vegetation in Lianmuqin Town, promotes the sustainable development of agriculture and animal husbandry, and improves the ecological environment of the water area. In the future, we should continue to optimize the ecological water conveyance strategy and rationally adapt the land use structure to establish a harmonious coexistence of ecology, economy, and society.

## 4. Discussion

### 4.1. Feasibility analysis and prospects for research

This study focuses on the Ertanggou Basin, an arid region in Xinjiang, analyzing discharge at four cross-sections within the basin. Results reveal that areas reliant on glacial meltwater as their primary source of water exhibit annual discharge directly influenced by temperature. High-altitude regions experience minimal water loss, resulting in substantial annual discharge. However, as elevation decreases, surface discharge velocity slows, water flow diverges, and a portion infiltrates into the ground, leading to reduced water volume downstream. In the downstream oasis areas, characterized by dense populations and frequent agricultural activities, significant amounts of water are diverted for irrigation, industrial,

and domestic use, further depleting reserves. This study does not account for interannual variations in precipitation or its impact on assessing river discharge and ecological water conveyance benefits. This omission is justified, as increased precipitation, while increasing available water in the basin, is not a critical factor for the arid Ertanggou Basin. Instead, the quantity of ecological water conveyance plays a decisive role in influencing downstream discharge. Agricultural development in the irrigation areas remains reliant on artificial water diversion, while oases are primarily distributed around rivers and low-lying regions where groundwater naturally seeps out [27,28].

Additionally, this study emphasizes discharge change patterns and the impacts of ecological water conveyance on land-use areas downstream but lacks a detailed analysis of the various factors contributing to these impacts. Consequently, the research does not sufficiently quantify the contribution rates of individual factors to discharge variations. Future research will address this gap by employing hydrological models [29], altering temporal and spatial scales [30,31], and analyzing changes in river morphology.

## 4.2. Comparative analysis of research outcomes

Numerous domestic and international scholars have employed diverse methods to investigate discharge changes across various river basins, yielding significant findings. Many studies have demonstrated that ecological water conveyance can provide additional water sources for rivers, enhancing the economic benefits of downstream agriculture. For example, Ailike SiLaiyin et al. [32] calculated environmental quality indices to comprehensively evaluate the natural environmental quality of the lower Tarim River. Their findings show that from 2000 to 2020, ecological water conveyance significantly improved the natural environmental quality of the lower Tarim River. Similarly, Wang Chuan et al. [33] used long-term monitoring and remote sensing data in the Ejina Oasis to analyze hydro-ecological indicators' responses to ecological water conveyance volumes, clarifying the driving factors and intensities influencing vegetation restoration in the oasis. Xinchen Gu et al. [34] constructed a SWAT hydrological model for the runoff-producing area of the Manas River Basin, analyzed the changes in hydrological elements, and predicted the future trend of water resources based on the CMIP5 model, indicating that in the Northwest Arid Zone, the future fluctuation range of water resources is large, and it is necessary to strengthen the monitoring and management of water resources. Unlike these previous studies, particularly when compared to large-scale water conveyance projects in China designed to alleviate water scarcity in northern regions [35,36], the Ertanggou Watershed project is specifically tailored to the ecological requirements of arid regions. This project emphasizes not only water resource storage but also integrates water resource protection and rational allocation to maximize ecological benefits. It underscores the importance of localized solutions when addressing ecological water challenges.

During the implementation of the Ertanggou project, the environmental impacts of reservoir construction and the long-term sustainability of the ecological water conveyance efforts were carefully considered [37,38]. Measures such as the construction of fish ladders and the preservation of wetland areas were adopted to minimize environmental disruption [39]. As more regions worldwide face water scarcity, the experience gained from the Ertanggou Watershed project offers valuable insights for developing ecological water conveyance strategies in other arid regions.

## 4.3. Mechanisms driving changes in landscape patterns by ecological water conveyance

Ecological water conveyance and its modification directly affect land use and coverage. This, in turn, impacts discharge distribution and creates an interactive relationship [40]. In arid and semi-arid regions, natural precipitation alone is insufficient to sustain the growth of forests and grasslands. Increasing ecological water conveyance enhances water availability, which improves vegetation growth and expands grassland coverage. Adequate water resources also support crop growth and promote agricultural development. Changes in forest and grassland areas feedback into the hydrological cycle, affecting the efficiency and demand for ecological water conveyance. For instance, expanding forest and grassland areas may improve the soil's water retention capacity, potentially reducing the need for water conveyance. Conversely, an increase in arable land may elevate irrigation water demand [41]. Analyzing the spatiotemporal dynamics of ecological water diversion

in the Ertanggou Watershed reveals an overall increasing trend in ecological water diversion from 2016 to 2020. However, pronounced imbalances in seasonal and spatial distribution persist. These dynamics are influenced by climate change, ecological water demand patterns, and long-term changes in glacier mass and meltwater rates. This is consistent with the research conclusions of other scholars at present. Among them, Zhang Hui et al. [42] analyzed long-term observational data from the Kuitun River Basin in the Tianshan Mountains of Xinjiang and found that over the past 20 years, glacier mass in the basin has steadily decreased, and the melt rate has accelerated, resulting in increased summer discharge. In contrast, winter discharge has decreased due to a reduction in glacial meltwater. This seasonal variation in glacier melt directly influences the spatiotemporal dynamics of ecological water conveyance. Similarly, Li ZH et al. [43] examined the discharge response of the Loess Plateau in China to environmental changes and found that, due to the long-term decline in glacier mass, the timing of peak flow has shifted. The traditional summer peak has occurred earlier, while winter flows have decreased, further exacerbating the imbalance in water distribution between upstream and downstream areas. The spatial variability in ecological water diversion underscores the complexity and regional differences inherent in the process. This suggests the necessity of optimizing water diversion strategies based on the ecological needs of different seasons and regions to ensure ecosystem stability and sustainable development [44]. Maintaining a stable forest and grassland landscape while enhancing the watershed's water conservation capacity is also critical to sustaining the ecological water sources of the basin [45]. Future research on ecological water diversion strategies should study and predict the long-term impact of diminishing glacier mass and changing meltwater rates to adapt to these evolving challenges [46]. Water diversion strategies must be adjusted to account for shifts in peak flow periods, ensuring that ecological water needs are met throughout different seasons. Strengthening the balance of water resource distribution between upstream and downstream areas through storage and regulation facilities, coupled with advanced technologies to optimize spatial water distribution, will also be essential. Furthermore, improving the watershed's water conservation capacity by protecting and restoring forest and grass vegetation can enhance the basin's water resource regulation functions, providing a scientific basis for sustainable water resource management [47].

## 4.4. Enlightenment of ecological construction in the Ertanggou watershed

The Ertanggou Watershed in Shanshan County, Turpan, Xinjiang, is an essential inland river watershed that plays a crucial role in the local social and economic development. The population in the watershed is mainly concentrated in the downstream oasis area, and the water supply relies on precipitation in the mountains and meltwater from snow and ice. Managing and utilizing limited water resources to meet ecological and agricultural needs has been a significant challenge for the local government. To address this, the government-initiated water conservation construction, starting with the reservoir project 2012, which was completed and accepted in November 2024. However, as agricultural land expands, the demand for water remains high, while surface water storage remains limited. Ecological water conveyance plays a crucial role in improving the ecological environment of arid and semi-arid regions. It helps restore wetland and river ecosystems, enhances soil fertility, and promotes biodiversity conservation while supporting the natural filtration of pollutants. However, ecological water conveyance also presents challenges, such as potential water quality issues, ecological disturbances, soil salinization, and competition for limited water resources. To maximize the benefits of ecological water conveyance while mitigating its negative impacts, several solutions must be implemented. First, strengthening water quality monitoring and treatment is essential to ensure that water quality meets environmental standards and reduces ecological pollution. Second, the design of water conveyance systems should be optimized to mimic natural water systems, minimizing disruption to the existing ecological balance. Additionally, soil fertility and salinization management should be prioritized, with measures such as rotational irrigation and intermittent irrigation to prevent soil salinization and enhance soil resilience. Natural filtration systems, such as wetlands, can be used in combination with agricultural management practices to manage pollutants and nutrients, thus reducing pollution sources. Finally, in response to climate change, flexible water conveyance plans should be developed to enhance the resilience and recovery capacity of ecosystems.

Therefore, it is essential to consider the hydrological effects of the growth process and the evolution of the landscape's spatial organization in promoting the ecological construction of the watershed. In the future, while supporting rural development, the primary focus should be the comprehensive management of environmentally sound small watersheds. Local governments should prioritize sensible distribution and exploitation of limited water resources to ensure ecological and agricultural needs rather than focusing solely on increasing land coverage. Achieving the sustainable development of the Ertanggou Watershed requires striking a balance between economic growth and environmental protection.

## 5. Conclusions

In this study, we focus on the Ertanggou Watershed. We used UAV and satellite remote sensing data to calculate the monthly discharge in the watershed from 2000 to 2020. We also evaluated the impact of ecological water conveyance from 2016 to 2020 and analyzed changes in the landscape pattern near the watershed, considering land cover types. The results show that:

(1) The RMSE and NSE river discharge values computed by RSHS technology are 0.94 and 3.2 $m^3$/s, respectively. This indicates that the discharge estimation method based on UAV and satellite remote sensing can provide valuable data for flow monitoring in the Ertanggou Watershed.

(2) From 2000 to 2020, the river discharge of the Ertanggou basin in Xinjiang grew overall. From 2016 to 2020, the discharge of the K1 portion upstream of the reservoir was much higher than that of the other sections, yet the monthly discharges of the K2, K3, and K4 sections downstream tended to be synchronized during the same period. This is primarily driven by river enhancement and the interception effect, significantly enhancing water resource efficiency and decreasing waste.

(3) From 2016 to 2020, the average annual ecological water conveyance in the Ertanggou basin was $0.61 \times 10^8$ $m^3$, with an efficiency of 10%, 2% higher than the previous 15 years. On the premise that river training and confinement projects significantly reduce losses along the ecological water conveyance, the ecological water conveyance can provide sufficient water resources for the downstream wetland as well as the downstream river perimeter.

(4) Land use change in the Ertanggou Watershed from 2000 to 2020 is characterized by increased built-up land and cropland and a decline in woodland. Economic development and the construction program of water conservancy facilities have resulted in considerable transfers of unused land, and the construction of reservoirs directly impacts the expansion of water bodies. The implementation of ecological water diversion has not only improved the ecological environment of the river basin but also promoted agricultural development by enhancing the efficiency of water resource utilization.

## Author contributions

**Data curation:** Junqing Lei, Zhanbo Wang.

**Funding acquisition:** Chao Xu.

**Investigation:** Zhanbo Wang.

**Project administration:** Adilai Wufu.

**Resources:** Adilai Wufu.

**Supervision:** Chao Xu.

**Writing – original draft:** Junqing Lei.

**Writing – review & editing:** Adilai Wufu, Chao Xu.

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
