## [Decision Letter · Decision Letter 0]

19 Nov 2024

PONE-D-24-49082Estimation of Long-Term River Discharge and Benefits Evaluation of Ecological Water Conveyance in Ertanggou BasinPLOS ONE

Dear Dr. Wufu,

Thank you for submitting your manuscript to PLOS ONE. After careful consideration, we feel that it has merit but does not fully meet PLOS ONE’s publication criteria as it currently stands. Therefore, we invite you to submit a revised version of the manuscript that addresses the points raised during the review process.

We look forward to receiving your revised manuscript.

Kind regards,

Vinaya Satyawan Tari, Post doctoral fellow, (M.Sc., B.Ed., Ph.D.)

Academic Editor

PLOS ONE

Journal Requirements: When submitting your revision, we need you to address these additional requirements. 1. Please ensure that your manuscript meets PLOS ONE's style requirements, including those for file naming. The PLOS ONE style templates can be found at https://journals.plos.org/plosone/s/file?id=wjVg/PLOSOne_formatting_sample_main_body.pdf and https://journals.plos.org/plosone/s/file?id=ba62/PLOSOne_formatting_sample_title_authors_affiliations.pdf 2. Thank you for stating the following financial disclosure: "This work was supported by the "Tianchi Yingcai" Talent Introduction Program of the Autonomous Region Talent Development Fund." Please state what role the funders took in the study.  If the funders had no role, please state: ""The funders had no role in study design, data collection and analysis, decision to publish, or preparation of the manuscript."" If this statement is not correct you must amend it as needed. Please include this amended Role of Funder statement in your cover letter; we will change the online submission form on your behalf. 3. We note that your Data Availability Statement is currently as follows: All relevant data are within the manuscript and its Supporting Information files Please confirm at this time whether or not your submission contains all raw data required to replicate the results of your study. Authors must share the “minimal data set” for their submission. PLOS defines the minimal data set to consist of the data required to replicate all study findings reported in the article, as well as related metadata and methods (https://journals.plos.org/plosone/s/data-availability#loc-minimal-data-set-definition). For example, authors should submit the following data: - The values behind the means, standard deviations and other measures reported;- The values used to build graphs;- The points extracted from images for analysis. Authors do not need to submit their entire data set if only a portion of the data was used in the reported study. If your submission does not contain these data, please either upload them as Supporting Information files or deposit them to a stable, public repository and provide us with the relevant URLs, DOIs, or accession numbers. For a list of recommended repositories, please see https://journals.plos.org/plosone/s/recommended-repositories. If there are ethical or legal restrictions on sharing a de-identified data set, please explain them in detail (e.g., data contain potentially sensitive information, data are owned by a third-party organization, etc.) and who has imposed them (e.g., an ethics committee). Please also provide contact information for a data access committee, ethics committee, or other institutional body to which data requests may be sent. If data are owned by a third party, please indicate how others may request data access. 4. PLOS requires an ORCID iD for the corresponding author in Editorial Manager on papers submitted after December 6th, 2016. Please ensure that you have an ORCID iD and that it is validated in Editorial Manager. To do this, go to ‘Update my Information’ (in the upper left-hand corner of the main menu), and click on the Fetch/Validate link next to the ORCID field. This will take you to the ORCID site and allow you to create a new iD or authenticate a pre-existing iD in Editorial Manager.

Reviewers' comments:

Reviewer's Responses to Questions

**Comments to the Author**

1. Is the manuscript technically sound, and do the data support the conclusions?

Reviewer #1: Partly

Reviewer #2: Partly

2. Has the statistical analysis been performed appropriately and rigorously? 

Reviewer #1: No

Reviewer #2: I Don't Know

3. Have the authors made all data underlying the findings in their manuscript fully available?

Reviewer #1: No

Reviewer #2: Yes

4. Is the manuscript presented in an intelligible fashion and written in standard English?

Reviewer #1: Yes

Reviewer #2: Yes

5. Review Comments to the Author

Reviewer #1: In this study, they focus on the Ertanggou Watershed, they have used UAV and satellite remote sensing data to calculate the monthly discharge in the watershed from 2000 to 2020. They have also evaluated the impact of ecological water conveyance from 2016 to 2020.

They have observed that the discharge estimation method based on UAV and satellite remote sensing can provide valuable data for flow monitoring in the Ertanggou Watershed. But they should also use the observed field discharge from secondary sources to make sure about the actual discharge happening in the region that is missing

From 2000 to 2020, the river discharge of the Ertanggou basin in Xinjiang grew overall. From 2016 to 2020, the discharge of the K1 portion upstream of the reservoir was much higher than that of the other sections, primarily driven by river enhancement and the interception effect, that significantly enhance water resource efficiency and decrease waste. But in previous years the discharge remains same why it is so? Has not been explained and also discharge peaked only in 2 years and again it has come back to the same level what is it meant for glacier melt that is linked to CC .In other location as well even though discharge is less as expected there is no variation in last two decades what may be the reason , they have simplified their observation it sems and it needs intense discussion here .Similarly , the average annual ecological water conveyance was 0.61×108m3, with an efficiency of 10%, 2% higher than the previous 15 years. What is the justification/reason for this change ;since no drastic discharge changes has-been observed after 2010 before 2010 that has occurred ,but it is not correlating with discharge changes in subsequent -the linkage between each parameter and their statistical significance has not been discussed clearly and it needs further discussion and revision

The annual or seasonal discharge rates are also to be discussed here that are influenced by precipitation, snowmelt, groundwater contributions, human activity, and of course climate change. Details of intervention affecting the discharge like hydropower plants, reservoir and dam etc., are to be linked in the discussion .

The Climate Change influenced by environmental aspects are also to be analysed and linked using hydrological modeling, ecosystem changes, change in sediment transport and river morphology, impact of agriculture actives ,etc .Since they are taking about the environmental flow impacts on nutrient loading for nutrient regulations is very important that has also to be brought out here as well.

The benefits and drawbacks of this ecological water conveyance are critical and are not addressed which will affect the soil fertility, biodiversity preservation ,habitat maintenance , natural filtration of pollutants and nutrients and adaptation to climate change.

They should bring out how this will help to mitigate these changes that are happening that will affect the ecosystems resilience that are altered to changes in flow conditions. This will reduce/enhance the vulnerability to climate impacts by balancing human and ecological needs of river water .Next from their observation can they bring out how this will help to make decisions on how much water can be safely extracted for human use without damaging ecosystems and for sharing by multiple users and enhance input for river and wetland restoration and other projects. They have not brought out the linkage between the long term changes impact of glacier mass and melt rate, shift in peak discharge periods and timing and resulted long-term decline in glacier mass and water availability in future.

Reviewer #2: Add the background of the study in the abstract.

It is necessary to specify what the project is for ecological water conveyance in the study area, as many international readers are not aware of the context of the study, consider placing it in the introduction or in the overview of the study map.

Explanations of formulas should be topspaced and not indented. Revise line 149 and its counterpart in the manuscript.

The results in Figure 3 are interesting and should be discussed in more detail.

The font in Figure 7 is too small and somewhat blurred.

The discussion is too brief and needs to be added. There should be a discussion of the research methodology, the similarities and differences between the results and others. For example, the prospect of the application of this technical method in the arid zone in comparison with other studies, whether there are ecological water transfer projects in other regions and their comparison, and other aspects.

Overall, the research is solid, but the writing of the manuscript should continue to improve. In particular, the title of the manuscript is based on the study of ecological water transfer, for readers in other regions do not understand what period of time the project began to realize its benefits, and then there is some lack of understanding of the assessment of its benefits.

6. PLOS authors have the option to publish the peer review history of their article (what does this mean? ). If published, this will include your full peer review and any attached files.

**Do you want your identity to be public for this peer review?** For information about this choice, including consent withdrawal, please see our Privacy Policy .

Reviewer #1: **Yes: ** A L Ramanathan

Reviewer #2: No

---

## [Author Response · Author response to Decision Letter 1]

1 Feb 2025

Thank you for your letter and the constructive comments on this article in your busy schedule. All of us authors have carefully read the comments that you have given us, and have discussed and revised each of these issues. In addition, we have resubmitted a new manuscript in the revised state, with the revisions highlighted in red. If there are any incorrect answers or questions in the manuscript, please do not hesitate to let us know.

---

## [Decision Letter · Decision Letter 1]

27 Feb 2025

PONE-D-24-49082R1Estimation of Long-Term River Discharge and Benefits Evaluation of Ecological Water Conveyance in Ertanggou Basin from 2000 to 2020PLOS ONE

Dear Dr. Wufu,

Thank you for submitting your manuscript to PLOS ONE. After careful consideration, we feel that it has merit but does not fully meet PLOS ONE’s publication criteria as it currently stands. Therefore, we invite you to submit a revised version of the manuscript that addresses the points raised during the review process.

We look forward to receiving your revised manuscript.

Kind regards,

Vinaya Satyawan Tari, Post doctoral fellow, (M.Sc., B.Ed., Ph.D.)

Academic Editor

PLOS ONE

Journal Requirements:

Additional Editor Comments:

Dear Authors,

Greetings!

This manuscript has been adequately improved. However, as per reviewer 2 some minor corrections are needed. Hence, pushed for revision 2.

Thanks and good luck

Reviewers' comments:

Reviewer's Responses to Questions

**Comments to the Author**

1. If the authors have adequately addressed your comments raised in a previous round of review and you feel that this manuscript is now acceptable for publication, you may indicate that here to bypass the “Comments to the Author” section, enter your conflict of interest statement in the “Confidential to Editor” section, and submit your "Accept" recommendation.

Reviewer #1: All comments have been addressed

Reviewer #2: All comments have been addressed

2. Is the manuscript technically sound, and do the data support the conclusions?

Reviewer #1: Yes

Reviewer #2: Yes

3. Has the statistical analysis been performed appropriately and rigorously? 

Reviewer #1: Yes

Reviewer #2: I Don't Know

4. Have the authors made all data underlying the findings in their manuscript fully available?

Reviewer #1: No

Reviewer #2: Yes

5. Is the manuscript presented in an intelligible fashion and written in standard English?

Reviewer #1: No

Reviewer #2: Yes

6. Review Comments to the Author

Reviewer #1: they have addressed most of all the queries raised by the reviewers by some are not answered clearly like what period of time the project began to realize its benefits, and then there is some lack of understanding of the assessment of its benefits. it may be ignored

Reviewer #2: Table 1, GEE is generally capitalized. Line 169, where initials need to be lowercase here. The formatting of references needs to be improved. The methods section could consider adding citations (https://doi.org/10.1007/s13201-023-02099-7).

7. PLOS authors have the option to publish the peer review history of their article (what does this mean? ). If published, this will include your full peer review and any attached files.

**Do you want your identity to be public for this peer review?** For information about this choice, including consent withdrawal, please see our Privacy Policy .

Reviewer #1: **Yes: ** AL Ramanathan

Reviewer #2: No

---

## [Author Response · Author response to Decision Letter 2]

6 Mar 2025

Response letter to Reviewer 1

Thanks a lot for your time and effort to improve our manuscript. The authors really appreciate your thoughtful and favorable opinions on our manuscript. We have considered carefully about your detailed comments, questions and opinions, and the manuscript was revised accordingly.

Comment: They have addressed most of all the queries raised by the reviewers by some are not answered clearly like what period of time the project began to realize its benefits, and then there is some lack of understanding of the assessment of its benefits. it may be ignored.

Answer: We appreciate your thorough review of our manuscript and the meticulous attention to detail in your comments. In response to your queries, we would like to provide the following clarifications:

Regarding the timeline of project benefits, this information is included in the study area overview. The project milestones are as follows: reservoir construction began in 2012, operational storage commenced in 2014, a regional water resources allocation feasibility meeting was held in 2015, and river regulation works were completed in 2016. These milestones outline the timeframe within which the project began yielding benefits.

The related content is as follows:

Line 123-134: To address the pressing demands of economic and social development and to optimize the allocation of water resources within the basin, the government completed the “Feasibility Study Report on Key Water Resource Allocation Projects in Shanshan County” in 2015. Following this, a series of water-saving transformation measures were implemented. Among these efforts, the construction of the Ertanggou Reservoir began in September 2012. By November 2014, the reservoir was operational, with a total storage capacity of 2.36×108 m3 and an irrigation control area extending to 202,600 acres. This reservoir has played a crucial role in alleviating severe water shortages in downstream oasis communities, increasing irrigation water for agricultural fields, and providing a reliable water resource supply for the daily needs of downstream urban residents [18]. By August 2016, the Ertanggou Reservoir had reached its normal storage level.

18.Zhao Fuyong. Hydrological calculation and analysis of the Ertanggou River in Xinjiang under engineering regulation [J]. Water Conservancy Science and Cold Region Engineering, 2021, 4(04): 139-142.

Regarding the assessment of the project's benefits, we have analyzed the achievements of the Ertanggou project in the comparative analysis section, emphasizing its role in water resource allocation and protection. Additionally, we have incorporated a summary in the conclusion, as per your suggestion, quantifying the benefits of ecological water conveyance. Our findings indicate that the project has enhanced both the volume and efficiency of ecological water conveyance in the Ertanggou River Basin, leading to an increase in water surface area and agricultural land, thereby demonstrating its positive outcomes.

We have carefully considered your feedback and made the necessary revisions to improve the clarity and robustness of our manuscript.

The related content is as follows:

Line 499-513: Unlike these previous studies, particularly when compared to large-scale water conveyance projects in China designed to alleviate water scarcity in northern regions [35-36], the Ertanggou Watershed project is specifically tailored to the ecological requirements of arid regions. This project emphasizes not only water resource storage but also integrates water resource protection and rational allocation to maximize ecological benefits. It underscores the importance of localized solutions when addressing ecological water challenges. During the implementation of the Ertanggou project, the environmental impacts of reservoir construction and the long-term sustainability of the ecological water conveyance efforts were carefully considered [37-38]. Measures such as the construction of fish ladders and the preservation of wetland areas were adopted to minimize environmental disruption [39]. As more regions worldwide face water scarcity, the experience gained from the Ertanggou Watershed project offers valuable insights for developing ecological water conveyance strategies in other arid regions.

35.Yan, H., Lin, Y., Chen, Q., et al. (2023). A Review of the Eco-Environmental Impacts of the South-to-North Water Diversion: Implications for Interbasin Water Transfers. Engineering, 30, 161-169. https://doi.org/10.1016/j.eng.2023.05.012

36.Xu Zhangxing, Liu Siyuan, Li Yiwen, Jiang Zheng & Jin Yu. The Impact of the South-to-North Water Diversion Project on the Non-agriculturalization of Water Resources in the Receiving Area. South-to-North Water Diversion and Water Science and Technology (in Chinese and English) 1-16.

37.Wang Sifan, Xu Hui, Ren Yufeng, et al. Research Progress on the Impact of Reservoir Construction on River Water Temperature and Ecological Environment. Three Gorges Ecological Environment Monitoring 1-29.

38.Chen Nuo, Bao Dan & Zhu Zhilin. (2024). Discussion on the Ecological Environment Construction of Wang jia chong Reservoir Area in the Water Resources Allocation Project of Northern Hubei. Theoretical Research in Urban Construction (Electronic Edition) (13), 202-204. doi: 10.19569/j.cnki.cn119313/tu.202413067.

39.Li Jiang, Liu Ying, Wu Tao & Peng Zhaoxuan. (2020). Seventy Years of Reservoir Dam Construction Achievements in Xinjiang. Journal of the China Institute of Water Resources and Hydropower Research (05), 322-330. doi: 10.13244/j.cnki.jiwhr.20200501.

The supplementary content is as follows:

Line 617-620: On the premise that river training and confinement projects significantly reduce losses along the ecological water conveyance, the ecological water conveyance can provide sufficient water resources for the downstream wetland as well as the downstream river perimeter.

Line 625-628: The implementation of ecological water diversion has not only improved the ecological environment of the river basin but also promoted agricultural development by enhancing the efficiency of water resource utilization.

Response letter to Reviewer 2

Thanks a lot for your time and effort to improve our manuscript. The authors really appreciate your thoughtful and favorable opinions on our manuscript. We have considered carefully about your detailed comments, questions and opinions, and the manuscript was revised accordingly.

Please see our point-to-point responses below.

Comment1: Table 1, GEE is generally capitalized.

Answer1: The "Gee" in Table 1 has been changed to "GEE".

Comment2: Line 169, where initials need to be lowercase here.

Answer2: The first word in line 169, "Where" has been changed to "where".

Comment3: The formatting of references needs to be improved.

Answer3: Thank you for your observation regarding the formatting of the references. I have carefully reviewed and made the necessary adjustments to ensure that the references now adhere to the journal's formatting requirements.

Comment4: The methods section could consider adding citations. (https://doi.org/10.1007/s13201-023-02099-7)

Answer4: Regarding your suggestion to strengthen the methods section with additional citations, we have reviewed the reference you provided (https://doi.org/10.1007/s13201-023-02099-7) and recognize its value in analyzing runoff changes and predicting water resource trends. Given our paper’s structure, we have incorporated this reference in the discussion section, where we compare our findings with existing studies. We also intend to explore its insights further in future research to optimize our experimental design.

Thank you again for your valuable suggestions and support.

The supplementary content is as follows:

Line 494-499: Xinchen Gu et al. constructed a SWAT hydrological model for the runoff-producing area of the Manas River Basin, analysed the changes in hydrological elements, and predicted the future trend of water resources based on the CMIP5 model, indicating that in the Northwest Arid Zone, the future fluctuation range of water resources is large, and it is necessary to strengthen the monitoring and management of water resources.

34.Gu, X., Long, A., He, X. et al. Response of runoff to climate change in the Manas River Basin flow-producing area, Northwest China. Appl Water Sci 14, 43 (2024). https://doi.org/10.1007/s13201-023-02099-7

---

## [Decision Letter · Decision Letter 2]

16 Mar 2025

Estimation of Long-Term River Discharge and Benefits Evaluation of Ecological Water Conveyance in Ertanggou Basin from 2000 to 2020

PONE-D-24-49082R2

Dear Dr. Wufu,

We’re pleased to inform you that your manuscript has been judged scientifically suitable for publication and will be formally accepted for publication once it meets all outstanding technical requirements.

Kind regards,

Vinaya Satyawan Tari, Post doctoral fellow, (M.Sc., B.Ed., Ph.D.)

Academic Editor

PLOS ONE

Additional Editor Comments (optional):

Reviewers' comments:

Reviewer's Responses to Questions

**Comments to the Author**

1. If the authors have adequately addressed your comments raised in a previous round of review and you feel that this manuscript is now acceptable for publication, you may indicate that here to bypass the “Comments to the Author” section, enter your conflict of interest statement in the “Confidential to Editor” section, and submit your "Accept" recommendation.

Reviewer #1: All comments have been addressed

Reviewer #2: All comments have been addressed

2. Is the manuscript technically sound, and do the data support the conclusions?

Reviewer #1: Yes

Reviewer #2: Yes

3. Has the statistical analysis been performed appropriately and rigorously? 

Reviewer #1: Yes

Reviewer #2: Yes

4. Have the authors made all data underlying the findings in their manuscript fully available?

Reviewer #1: Yes

Reviewer #2: Yes

5. Is the manuscript presented in an intelligible fashion and written in standard English?

Reviewer #1: Yes

Reviewer #2: Yes

6. Review Comments to the Author

Reviewer #1: The authors have addressed almost all the comments of the referees in satisfactorly and hence it may be considered

for publications

Reviewer #2: (No Response)

7. PLOS authors have the option to publish the peer review history of their article (what does this mean? ). If published, this will include your full peer review and any attached files.

**Do you want your identity to be public for this peer review?** For information about this choice, including consent withdrawal, please see our Privacy Policy .

Reviewer #1: **Yes: ** AL Ramanathan

Reviewer #2: No

---

## [Editor Report · Acceptance letter]

PONE-D-24-49082R2

PLOS ONE

Dear Dr. Wufu,

I'm pleased to inform you that your manuscript has been deemed suitable for publication in PLOS ONE. Congratulations! Your manuscript is now being handed over to our production team.

Kind regards,

on behalf of

Dr. Vinaya Satyawan Tari

Academic Editor

PLOS ONE